# Microbial coexistence through chemical-mediated interactions

Lori Niehaus[1], Ian Boland[1], Minghao Liu[2], Kevin Chen[1], David Fu[1], Catherine Henckel[1], Kaitlin Chaung[1], Suyen Espinoza Miranda[1], Samantha Dyckman[1], Matthew Crum[1], Sandra Dedrick[1], Wenying Shou[3] & Babak Momeni[1]

Many microbial functions happen within communities of interacting species. Explaining how species with disparate growth rates can coexist is important for applications such as manipulating host-associated microbiota or engineering industrial communities. Here, we ask how microbes interacting through their chemical environment can achieve coexistence in a continuous growth setup (similar to an industrial bioreactor or gut microbiota) where external resources are being supplied. We formulate and experimentally constrain a model in which mediators of interactions (e.g. metabolites or waste-products) are explicitly incorporated. Our model highlights facilitation and self-restraint as interactions that contribute to coexistence, consistent with our intuition. When interactions are strong, we observe that coexistence is determined primarily by the topology of facilitation and inhibition influences not their strengths. Importantly, we show that consumption or degradation of chemical mediators moderates interaction strengths and promotes coexistence. Our results offer insights into how to build or restructure microbial communities of interest.

---

[1] Department of Biology, Boston College, Chestnut Hill, MA 02467, USA. [2] Department of Computer Science, Boston College, Chestnut Hill, MA 02467, USA. [3] Division of Basic Sciences, Fred Hutchinson Cancer Research Center, Seattle, WA 98109, USA. Correspondence and requests for materials should be addressed to B.M. (email: momeni@bc.edu)

Microbial communities impact ecosystems by cycling matter[1] and affect human health by assisting food digestion or causing infections[2–5]. Cohabiting microbes in communities interact and can perform functions that none of the member species can achieve efficiently on their own[6,7]. Examples of such functions include degradation of complex compounds such as crude oil, cellulose, or plastics[8–10] or resistance against pathogen colonization[11]. How can species in a community stably coexist, despite differences in their intrinsic growth rates? To design long-lasting communities for waste remedy or fuel production[12], or to manipulate host-associated communities[13], a better understanding of coexistence mechanisms will be instrumental.

Exploring what allows coexistence (defined as extended presence of different species within a community) and stability (defined as maintenance of coexistence despite perturbations) has been among major directions in community ecology[14–17]. Many studies, both theoretical and experimental, have identified how coexistence may be achieved[18]. Species interactions have been recognized as a means of achieving coexistence. Facilitation (i.e., influences that benefit one of the partners), for example, has been identified to support coexistence, by boosting the growth of intrinsically less fit recipients[19,20], relieving facilitators from competitive pressure[19], or protecting vulnerable species from harsh environments or predators[20,21].

Most previous models of communities abstract all the interactions between species into pairwise fitness interactions[22–26]. This simplification is intended to recapitulate how each interaction influences the fitness of the two involved parties[14,22,23,27]. However, pairwise fitness models may not accurately capture common situations in which interactions take place through different mechanisms (e.g., via a consumable metabolite or a change in the environment) or when shared mediator is produced or consumed by multiple species[28–30].

Here, we use a mediator-explicit model which incorporates chemical mediators of interactions[28]. This choice is motivated by several considerations: (1) Interactions mediated by chemical compounds (e.g., metabolites or toxins) are common among microbes[5,31,32], and are thought to be influential. (2) Indirect interactions where one species affects how strongly other species interact[33–35] are lost in pairwise interaction models but preserved in mediator-explicit models. (3) Recent progress in stable isotope probing (SIP)[36], mass spectroscopy (MS)[37], and nuclear magnetic resonance (NMR)[38] has improved our ability to identify and quantify interaction mediators. In our model, we focus on how interactions lead to coexistence when external nutrients are replenished to be in excess and species growth rates are modulated by chemical compounds produced and consumed by cells. Continuous growth can be found in environments such as turbidostats, some industrial bioreactors, or possibly human gut in which resources are continuously or cyclically supplied[39]. This model can be considered as a special type of consumer-resource model[29,30,40] in which chemical mediators generated by species are modeled, but external resources are not modeled since they are supplied in excess.

Our results show that facilitation (i.e., growth-promoting influences) and self-restraint (i.e., inhibition of self) contribute to coexistence, as intuitively expected. We observe that facilitation and self-restraint often causally support coexistence, whereas inhibition of other species is detrimental to coexistence. When interactions are strong, the topology of facilitation and inhibition influences through chemical mediators appears to shape coexistence. We also show that interactions through depletable mediators (i.e., mediators consumed/degraded by recipients) are more conducive to coexistence compared to interactions through reusable mediators (i.e., mediators unaffected by recipients).

## Results

**Constructing a mediator-explicit model of interactions.** In our model, species interact with other members of the community through chemical mediators (Fig. 1a)[28]. Each species can produce multiple chemicals and each chemical can influence multiple species (Fig. 1b). To clarify our nomenclature, interactions are how species impact the growth rate of their own type (intraspecies) or other species (interspecies). Each of these interactions might be the result of multiple chemical influences, or influences for short. Each influence in our model represents how a chemical produced by community members affects the growth rate of a species. In our network, we indicate growth rate influences on species (from a chemical to a species) as f-links, and chemical production/removal by species as c-links. A link may refer to a c-link or an f-link. Even though microbes are expected to change many chemicals in their environment[37], in our model we consider a finite number of mediators (typically 5–20). Our reasoning is that (1) we are only including mediators with strong growth rate influence, and (2) as a simplification, different mediators may be grouped into functional categories, such as organic acids or small sugars, simplifying community models[41].

To build the model based on realistic assumptions, we assessed examples of how chemical mediators affect the growth of cells. We experimentally characterized the growth of bacterial cells in the presence of different concentrations of chemical compounds, $C_l$, that stimulated or inhibited growth (Fig. 1c, d). For growth inhibitors (Fig. 1c), we have frequently observed that the growth rate linearly drops as the concentration of the inhibitor increases (Fig. 1c and Supplementary Fig. 1). For inhibition by antibiotics, we have observed that the inhibition is typically exerted above a threshold concentration (Supplementary Fig. 1). A previous report suggests another possible form of inhibition with a threshold effect at low inhibitor concentrations and a reduced impact of inhibition at high inhibitor concentrations[42] (see Methods). For simplicity, we choose the form in which growth rate linearly drops as inhibitor concentration increases. We will show later that our results are not sensitive to this choice (Supplementary Fig. 6). For growth facilitators, we observe the common biological situation in which over-abundance of the mediator does not proportionally contribute to the growth rate (Fig. 1d). Our work (Fig. 1d and Supplementary Fig. 2) and others'[43] suggest that a general saturating form, Moser equation $C_l^n/(C_l^n + K_{i,l}^n)$, provides a good approximation for many cases of response to growth facilitators. $K_{i,l}$ is the concentration that parametrizes the saturating form of the dependence on the chemical concentration. For simplicity, we adopt the Monod form $C_l/(C_l + K_{i,l})$ to model this saturating behavior (see Methods) and show that our simulations are insensitive to this simplification (Supplementary Fig. 6).

By representing chemical concentrations as $C_1, \ldots, C_M$, and live species cell densities as $S_1, \ldots, S_N$, changes in concentrations of chemicals and populations of species can be described in our modified model as

$$\begin{cases} \frac{dS_i}{dt} = \left[ r_{i0} + \sum_l \left( \rho_{il}^+ \frac{C_l}{C_l + K_{i,l}} - \rho_{il}^- \frac{C_l}{K_{i,l}} \right) \right] S_i \\ \frac{dC_l}{dt} = \sum_l \left( \beta_{li} S_i - \alpha_{li} \frac{C_l}{C_l + K_{i,l}} S_i \right) \end{cases} \quad (1)$$

where $\alpha_{li}$ is the maximum rate of consumption of $C_l$ per $S_i$ cell, $\beta_{li}$ is the rate of production of $C_l$ per $S_i$ cell, $r_{i0}$ is the net basal growth rate of $S_i$ in the absence of chemical-mediated interactions, and $\rho_{il}^+$ (if positive) and $\rho_{il}^-$ (if negative) represent the influence of $C_l$ on the growth rate of $S_i$. Even though $\rho_{il}^-$ and $K_{i,l}$ can be collapsed into a single term, we have chosen to use the current form so that we can directly compare $\rho_{il}^-$ with $\rho_{il}^+$. The death rate in this

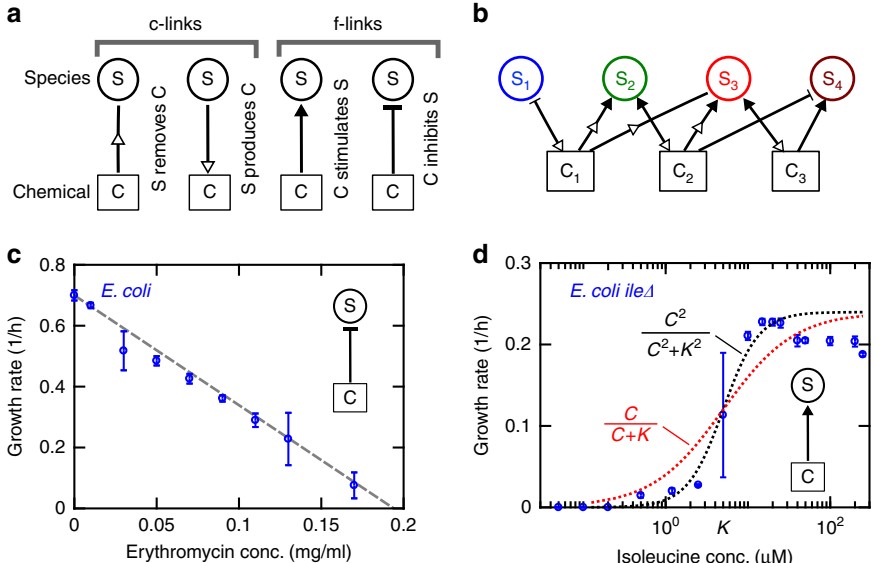

**Fig. 1** By explicitly incorporating the chemicals, we simulate a community of microbes engaged in chemical-mediated interactions. **a** Community networks are defined by two types of links: c-links (chemical production/removal links) indicated by hollow arrowheads and f-links (growth rate influences on species) indicated either by filled arrows for facilitation or bar-termination for inhibition. **b** A combination of c-links and f-links represents a community of species interacting through chemical mediators. **c** Inhibitory chemicals are assumed to linearly decrease the growth rate of species (see Methods), motivated by experimental measurements of growth in the presence of antibiotics and metabolic byproducts (Supplementary Fig. 1). **d** The influence of a facilitative chemical compound on species is approximated by the Monod equation for simplicity. Experimental observations of auxotrophic *E. coli* K-12 strains suggest that a second-order Moser equation (black dotted line; see Methods) offers a more accurate estimation (Supplementary Fig. 2), but the Monod equation (red dotted curve) is still an acceptable approximation. Error bars show s.d. based on 6 technical replicates

formulation is absorbed into the net growth rate, $r_{i0}$, and only live cells (with densities $S_i$) contribute to removal and production of chemicals. The consumption/degradation of mediators is included in this formulation through consumption factors, $\alpha_{li}$. Note that consumption of mediators that facilitate the growth of other species can be conceptually treated as competition. Similarly, consumption of mediators that inhibit the growth of other species effectively provides a benefit to them. If a mediator is consumed/degraded by a cell, we call it a depletable mediator. In the special case of reusable mediator, cells are affected by the mediator without considerably consuming or degrading it (e.g., in response to a signaling molecule), and we set $\alpha_{li} = 0$. We have also examined the effect of mediator decay and observed that except in extreme conditions (e.g., very rapid decay of mediators), the results were not sensitive to mediator decay. The combined effect of multiple mediators on the growth rate of each recipient species is assumed to be additive, similar to McArthur's model of resource utilization[40].

This modeling platform is fairly general and can capture a variety of inhibitory and facilitative chemical interactions[44]. Such interactions can include for example the effect of pH (modeled as the concentration of $H^+$), which is known to impact community structure[45,46]. In what follows, we will use this model to examine how chemical interactions among microbes may allow different species to coexist (Fig. 2). In our analysis of coexistence, we will rely on the experimentally-motivated model formulation in Eq. (1). Nevertheless, we will show that our findings depend on the sign and strength of interactions (Fig. 3), but not on the details of this formulation (Supplementary Fig. 6).

**Interspecies interactions can lead to coexistence of species**. To obtain communities that exhibit species coexistence, we simulate cycles of growth and dilution to emulate a typical experimental setting[39,47–49] called enrichment[50,51]. We call the resulting communities derived communities. Since shared resources are

replenished cyclically to be in excess, cells are not limited by shared resources, but instead grow at a rate dictated by Eq. (1). Mediators produced/consumed by cells modulate the growth rate of species and determine community interactions and coexistence. This allows us to focus primarily on how intercellular interactions can contribute to species coexistence.

In each simulation, we initially put together several species at equal proportions with a random network of interactions. These communities grow (following Eq. (1)) from a set total initial cell density ($\Sigma S_{\text{init}}$) up to a pre-determined threshold cell density for dilution ($\Sigma S_{\text{dil}}$), upon which the culture is diluted back to the initial cell density (Fig. 2a, top). Therefore, species with larger growth within each round will be over-represented in the next round. This can be considered as competition for space in the inoculum for the next round, leading to the coexistence of species that overall (because of their basal growth rate and interactions exerted by other species) grow the fastest (Fig. 2a, bottom). In this setting, each species can grow on its own in the supplied shared resources in the absence of chemical mediated interactions. Growth is simulated for 200 generations, which is typically enough to reach a steady pattern of population dynamics within each cycle (Supplementary Fig. 3). In this process, species whose density drops below an extinction threshold are considered extinct and removed from the rest of the simulation.

We define coexistence based on species that persist in the process of enrichment. Choosing the definition of coexistence faces a tradeoff. If all species that are present within a time frame are considered to coexist, we will inevitably include species that will eventually go extinct beyond the time frame. On the other hand, if we include only instances that we can verify to truly show coexistence over a very long term, then we will deviate from experimental feasibility as such a verification is unlikely implemented. Thus, we have chosen a balance between these two extremes, adopting an experimentalist's point-of-view (for example, as in refs. [49,52]). Specifically, we have defined

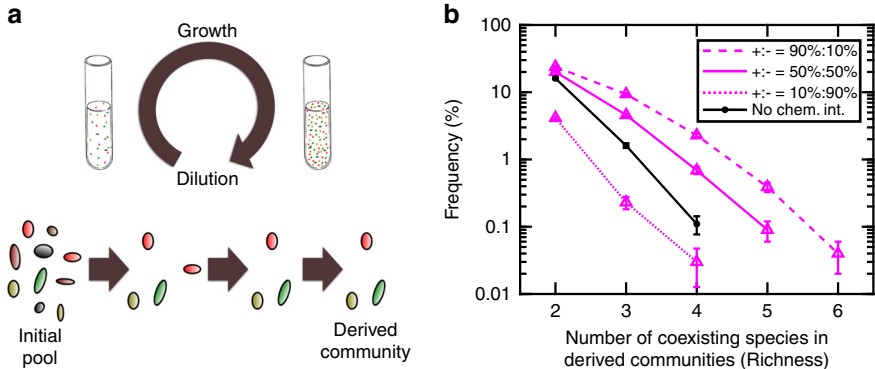

**Fig. 2** Simulated enrichment produces instances of species coexistence. **a** We simulate a typical experimental process of enrichment, in which a community of several species is grown in excess shared resource and is periodically diluted to a set density. In this process, we remove species that drop below a viable threshold abundance (corresponding to 1 cell in the inoculum). After several rounds of dilution (around 200 generations), we have observed that a subset of species remain in the community; this is considered an instance of coexistence (see Methods). **b** The likelihood of coexistence declines with community richness and increases with facilitative interactions. Community richness is defined as the number of coexisting species. As a reference, we have calculated the likelihood of observing coexistence for the same set of parameters, in the absence of any chemical interactions (solid black curve). In these simulations, we have assumed that chemicals that influence a species in half of the cases are depletable ($\alpha_{li} > 0$ in Eq. (1)), whereas in the other half, they are reusable ($\alpha_{li} = 0$ in Eq. (1)). We chose the initial number of species types $N_c = 20$ and the number of mediators $N_m = 15$. Error bars indicate s. d. due to sampling

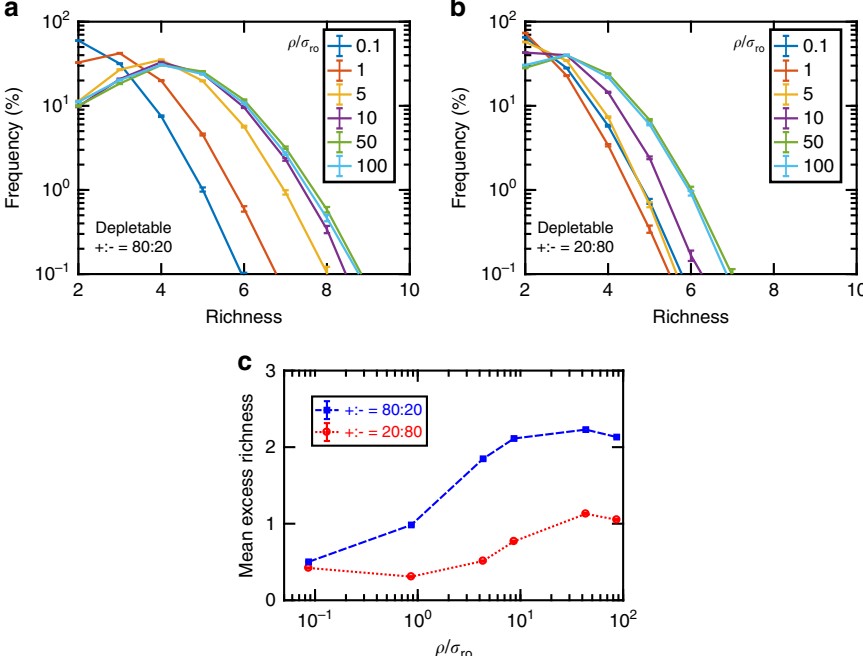

**Fig. 3** Coexistence is disrupted at weak influence strengths. Whether influences are mostly facilitative (**a**) or inhibitory (**b**), as interactions become stronger (i.e., larger $\rho/\sigma_{r0}$ values, where $\rho$ is the average influence strength and $\sigma_{r0}$ is the standard deviation of net basal growth rates in the initial pool), coexistence becomes more likely, but this trend saturates at strong influences. At very weak influence strengths, we can assume that coexistence happens only for species with very similar growth rates, approaching the neutral theory prediction in Fig. 2. Here the initial number of species types $N_c = 20$ and the number of possible mediators $N_m = 15$. Error bars indicate s.d. due to sampling. **c** Mean excess richness indicates average richness in derived communities beyond single-species dominance predicted by competitive exclusion principle (see Methods). Stronger interactions appear to contribute to more coexistence in a saturating form. Error bars show bootstrap estimates of 95% confidence intervals for the mean values

coexistence operationally as persistence of species in the community after a given amount of community growth (here, 200 generations). If the population size of a species drops below one cell, then the species is considered extinct and removed from the rest of the simulation. If the population size of a species drops by more than 10% in the final 20 generations of community growth, we also remove it from coexistent communities, because it will presumably slowly go extinct. This definition of coexistence is consistent with long-term stability in the sense that in most cases, if we had extended growth time without introducing perturbations, species would continue to coexist (Supplementary Fig. 4). Our definition of coexistence is also consistent with asymptotic stability in the sense that in majority of cases, if we perturb species frequency away from the steady state value, species frequency will return to the original steady state value (Supplementary Fig. 4).

Figure 2b shows the likelihood of reaching communities of different richness, starting from a pool of 20 randomly interacting species. In these simulations, we have assumed that the initial pool of species has a random connectivity network in which c-links and f-links each have a fixed presence–absence probability (i.e., each have a Erdős–Rényi connectivity graph[53]). We will call such bipartite networks binomial throughout this work. Under these assumptions, the model predicts that the likelihood of achieving communities with higher species richness decreases exponentially. As a control, we examined a community with similar parameters, but with no chemical interactions (representing the situation within the neutral theory of coexistence). In this situation, species with the highest fitness can coexist if their basal growth rate happens to be close enough, but the chance of coexistence is exponentially lower for communities with higher richness (Fig. 2b, black curve).

Comparing communities with different ratio of facilitation versus inhibition influences in the initial pool, we see that a community dominated by inhibitory influences has even lower chance of coexistence compared to the no-chemical-interaction control. As the ratio of facilitative versus inhibitory influences in the initial pool increases, the chance of coexistence also increases (Fig. 2b). This shows that the ratio of facilitation versus inhibition influences in the initial pool impacts coexistence, consistent with previous experimental observations[47,52].

Do members with the highest basal growth rate have a higher chance of surviving? In communities with no chemical interactions, that is indeed the case, but the pattern largely disappears in communities of species with strong chemical interactions (Supplementary Fig. 5). Coexistence in the presence of interactions does not favor species with the highest basal growth rate (Supplementary Fig. 5), suggesting that interactions are determining which species will survive in the simulated communities.

We use the flexibility of the mathematical model to explore how different properties of species and their interactions in the initial pool of microbes could influence the chance of coexistence. For simplicity, we change one parameter at a time and investigate the impact on the chance of achieving coexistence. The exact formulation of the facilitative and inhibitory influences does not appear to have a large impact on coexistence (Supplementary Fig. 6). We examined the outcome when influence strengths ($\rho_{il}$) had a distribution with a bias toward weak interactions. Our results suggest that enrichment outcome is not sensitive to the details of the distribution of influence strengths (Supplementary Fig. 7). We also find that the dilution scheme has only a modest influence on coexistence, with more strict dilutions leading to less coexistence (Supplementary Fig. 8), consistent with previous reports[54].

Our findings qualitatively match previous experimental observations: if the initial pool contains more inhibition influences, the chance of achieving coexistence through enrichment will be lower. We revisited three corresponding reports' experimental observations: Friedman et al. investigated coexistence among soil isolates[52]; Higgins et al. examined a larger set of pairwise interactions among soil isolates in a lab environment[48]; and Wright & Vetsigian examined cocultures of pairs of strains from the genus *Streptomyces*, observing mostly competitive exclusion and observing coexistence only at a low rate[47]. In ref. [52], coexistence was observed among most pairs studied (34 out of 56 pairs), suggesting that many species pairs might be engaged in facilitation[55]. Assuming this is the case, we would expect that many trios would also show coexistence, which is consistent with the observed results (19 out of 28 trios). In ref. [48], there are fewer instances of pairwise coexistence (19 out of 190 pairs) and more instances of bistability (15 out of 190 pairs), suggesting fewer facilitation and more inhibition among these

species, compared to ref. [52]. Interestingly, starting from a pool of all 20 species in ref. [48], the only trio that showed coexistence had species that all coexisted in pairs as well. This is consistent with the speculation that these three species facilitate each other's growth. In ref. [47], the strains in the initial pool are known to engage in inhibition (as evidenced by bistability in their cocultures), and the results showed even lower likelihood of coexistence (7 out of 153 pairs, after removing redundancies), consistent with our qualitative prediction. Because of the large variability in the experimental data, we speculate that coexistence is highly dependent on the strains placed in the initial pool (in turn determined by the ecological and evolutionary background of those strains). Detailed evaluation of expected coexistence requires a more thorough investigation of interactions among species being studied, which is beyond the scope of this work.

**Topology of the network of influences drives coexistence.** Considering the difference between cases with and without interactions in Fig. 2b, we asked how strong the interspecies interactions had to be to drive coexistence outcomes. We use $\rho/\sigma_{r0}$ as a measure of the strength of interactions: through interactions mediated by chemicals (with average strength $\rho$), species achieve coexistence by compensating the difference in their basal growth rates, quantified as the standard deviation of basal growth rates $\sigma_{r0}$. Our results show that indeed when all influences are chosen to be weak, coexistence is driven by neutral theory (Fig. 3, $\rho/\sigma_{r0} = 0.1$). We also observe that beyond some level, increasing the influence strength does not further favor coexistence (Fig. 3, $\rho/\sigma_{r0}$ above 10). This result suggests that when interactions are strong, the coexistence outcome may be determined by the qualitative network topology (e.g., who facilitates/inhibits whom), and not the quantitative influence strengths. When the interactions are strong, coexistence appears to be insensitive to the mean basal growth rates of species in the initial pool, but dependent on the standard deviation of basal growth rates (Supplementary Fig. 9). This is consistent with the intuition that interactions have to compensate for the differences in basal growth rates and a larger standard deviation makes coexistence less likely (with other parameters fixed). To quantify the level of coexistence in each case, we have defined mean excess richness (MER) as the average richness beyond single-species dominance predicted by competitive exclusion principle, in derived communities across all sampled initial pools (Methods). MER trends (Fig. 3c) confirm the saturating trends observed in Fig. 3a, b.

To further examine the impact of signs and strengths of influences on coexistence, we randomly changed the signs of a fraction of influences (Supplementary Fig. 10A) or modulated the influence strengths by a multiplicative factor (Supplementary Fig. 10B). We observe that the coexistence outcome is more sensitive to the sign of influences (transition from facilitation to inhibition and vice versa, even in a small subset of influences) compared to quantitative changes in interaction strengths (Supplementary Fig. 10).

**Facilitation and self-restraint are favored in enrichment.** Among communities that showed coexistence, we searched for shared features. We categorized chemical influences based on whether they were facilitative versus inhibitory, and whether the species affected themselves versus other community members. Influences thus belong to one of four categories: self-facilitation, other-facilitation, self-restraint, and other-inhibition. We asked how enrichment for coexistence favored or disfavored each of these categories of influences. To answer this question, we compared how the frequency of each influence category changed from the initial pool to the final derived community of coexisting

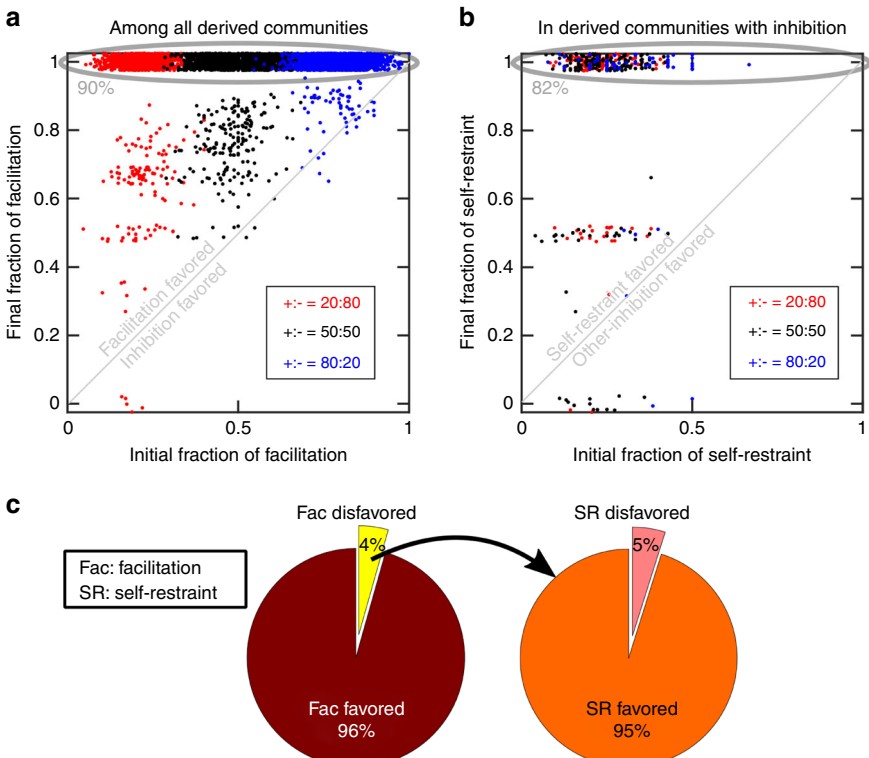

**Fig. 4** Facilitation and self-restraint are favored in enrichment. We examined how the fraction of interaction types changed from the initial pool to the final derived community. Each dot here represents an enrichment process going from 20 initial species to a final community with at least two coexistent species. We examined three conditions with the initial ratio of facilitation to inhibition influences being 20:80 (red), 50:50 (black), or 80:20 (blue). **a** Facilitation was highly favored (i.e., its frequency increased from the initial pool to the derived communities), and 90% of derived communities contained only facilitation. **b** In final communities that contained inhibition, self-restraint was prevalent (82% in this category contained only self-restraint), even though the fraction of self-restraint was only 20 ± 11% (s.d.) in the initial pool. Points above the diagonal line represent that facilitation and self-restraint are favored in (**a**) and (**b**), respectively. Fractions in derived communities appear quantized (especially in (**b**)) due to the small numbers of influences remaining (1 out of 1, 1 out of 2, 2 out of 2, 1 out of 3, etc.). The data points in (**a**) and (**b**) are jittered by 5% to reduce overlap for visualization purposes. **c** We examined the break-down of different categories of influences. Facilitation was favored in 96% of communities during enrichment, including in all communities that lacked self-restraint. Among the 4% in which facilitation was disfavored, in 95% of cases self-restraint was favored during enrichment. This suggests that facilitation is the main driver of coexistence, with self-restraint being a secondary means of achieving coexistence. In these simulations, the number of initial pools examined $N_s = 30,000$, the initial number of species types $N_c = 20$, and the number of possible mediators $N_m = 15$. All mediators in these simulations are depletable; the same trends hold when mediators are all reusable

species. Our results suggest that in derived communities, facilitation and self-restraint (i.e., production of chemicals that has an inhibitory effect on the producer) are favored (Fig. 4a, b). Facilitation appears to be prevalent in derived communities, even if they are rare in the initial pool of interacting species (Fig. 4a). This conclusion is general and holds regardless of the details of the parameters of the initial pool (e.g., when we vary the initial ratio of facilitative to inhibitory influences).

The conclusion that facilitation and self-restraint arise as features of communities with coexistence is not surprising. The explanation for facilitation is intuitive: if a facilitative species rises in frequency, it improves the growth of its cohabitants and thus promotes coexistence. It is also intuitive that the negative self-feedback through self-restraint could prevent a species from outcompeting other members: as that species becomes more dominant, so becomes the inhibition it exerts on itself. This internal feedback, even if applied only to a few dominant members, can be the balancing force that allows coexistence.

Facilitation and self-restraint both have been suggested to play a role in coexistence and stability. From simpler two-species communities, we know that facilitation plays an important role in coexistence[55]. Facilitation has also been implicated from field work on plant communities to increase community richness[56].

Self-restraint is typically intrinsically assumed in pairwise models of ecological networks as a negative diagonal term in the matrix of interactions to incorporate the effect of intra-population competition[57]. It is worth noting that in our analysis, the model was agnostic to this potential, yet self-restraint emerged as one of the features of derived communities that exhibited coexistence.

**Coexistence is built around facilitation and self-restraint.** Considering that facilitation and self-restraint were correlated with coexistence, we asked if the relationship was causal; i.e., do different influence categories impact coexistence differently? To answer this question, we performed in silico knock-out experiments in which we removed an influence link from the community to examine how it affected richness.

From the final derived community, we picked one influence (chemical influence on a species) at a time, removed that influence from the initial pool of species, and asked how species richness was affected as a result. Our analysis shows that removing facilitative influences often leads to derived communities with fewer species (Fig. 5a). The same general pattern holds when we explore initial pools with other parameter values. This shows that facilitation has a positive causal impact on coexistence.

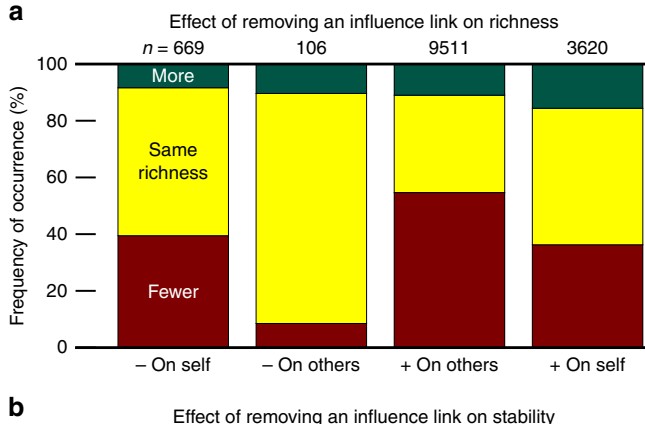

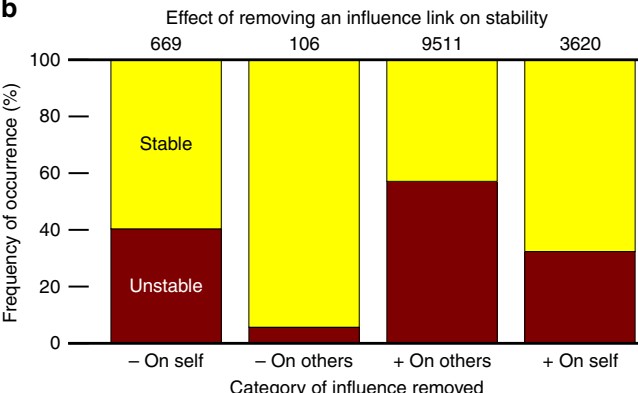

**Fig. 5** Even individual influences may causally impact coexistence and stability. In knock-out experiments we assess how removing an influence link from the community impacts coexistence. In (**a**), the influence is removed from the initial pool to ask how enrichment might be affected. Removing facilitation links has a considerable adverse effect on coexistence (with a large fraction of cases showing a drop in final richness). In contrast, removing self-restraint has a modest influence on coexistence, and removing inhibition of others on average benefits coexistence. In (**b**), the influence was removed from the final community to see how an already stably coexistent community was affected by removal of different types of links. Observations in (**b**) match the trends in (**a**). Removal of facilitation likely disrupts the stable community, whereas removal of inhibition of others is unlikely to impact the community. Self-restraint interactions are at intermediate importance; their removal has a modest (~17%) chance of disrupting an already stable community in this example. Here, the initial pool has a binomial network with equally likely positive or negative influences ($+:- = 50\%:50\%$). Since examined influences are chosen from derived communities that exhibited coexistence, more facilitative influences are present (e.g., Fig. 4) and thus tested ($n = 13,131$) than inhibitory influences ($n = 775$)

Among negative influences, self-restraint seems to contribute to coexistence more than other-inhibition, as removal of self-restraint more often leads to communities with lower richness (Fig. 5a).

We also started from the coexisting derived community, removed an influence, and asked if the community remained stable afterwards. We operationally define stability by testing if starting from a community that has exhibited coexistence, after removing an influence the same species coexist over the following 200 generations. We observed a trend similar to Fig. 5a, in which removal of other-inhibition is least likely to disrupt stability, removal of self-restraint has intermediate chance of making the community unstable, whereas removing facilitation influences has a high chance of disrupting stability (Fig. 5b). This general

observation seems to hold, regardless of the details of the parameters (Supplementary Fig. 11).

Our interpretation is that during enrichment, some influences will remain in the community even though they are not necessary for coexistence. It appears that facilitative influences (and to an intermediate degree, self-restraint influences) are the necessary links that maintain the coexistence of species. In contrast, the remaining influences, especially other-inhibition ones that are disruptive to coexistence, may hitchhike from the initial pool to the final derived community. In other words, a member species that is engaged in several facilitation interactions with other species may be able to tolerate one or more inhibitory influences. If such inhibitory influences are present in the initial pool, they might not prevent coexistence and thus persist to the derived communities. Removing these unnecessary influences is unlikely to make the community unstable.

**Coexistence is enhanced when mediators are consumed/degraded**. Earlier work had suggested that depending on whether chemical mediators are consumed/degraded (depletable mediators) or not (reusable mediators), qualitatively different dynamics are expected[28]. We asked whether this difference in interaction mechanism (i.e., whether or not mediators are removed from the environment by cells) impacted coexistence. We examined communities with the same network of connectivity, but varied the fraction of species that consumed/degraded the chemical mediator (depletable mediator). We observed that if a higher fraction of interactions take place through depletable mediators, coexistence becomes more likely (Fig. 6).

Considering that as the consumption/degradation rate increases, a mediator that is reusable can become depletable, we asked how coexistence depended on production and consumption rates. Specifically, we changed the ratio of the average rates of consumption to production of chemical mediators and monitored its impact on coexistence. With stronger consumption-to-release ratio (moving towards more depletable and fewer reusable mediators), more coexistence is achieved. This trend was more pronounced when there are more positive influences in the community and seems to saturate at very high ratios of consumption to production rates (Supplementary Fig. 12).

Our interpretation is that consumption acts as a negative feedback on species that receive facilitation from chemicals and have the potential to become dominant; this allows coexistence of more species. This interpretation is consistent with the observation that improved coexistence is missing when consumption is small or when most influences are inhibitory.

## Discussion

We used a model of microbial interactions mediated by chemicals to simulate microbial coexistence. Including chemicals that mediate the interactions between species has the potential to represent important features of microbial communities more accurately. Notably, recent work[29,30] has examined the incorporation of metabolites as resources to investigate coexistence. We examined a continuous growth situation (similar to refs. [39,49], where shared resources are being supplied), and incorporated production and consumption/degradation of mediators that can positively or negatively impact the growth of species within the community. We found that facilitation and self-restraint interactions played a critical role in allowing species coexistence. Importantly, removal of facilitation or self-restraint influences negatively impacted coexistence and stability. We also examined the effect of different parameters and showed that the prevalence of facilitation and the consumption/degradation of mediators are among major factors that impact coexistence.

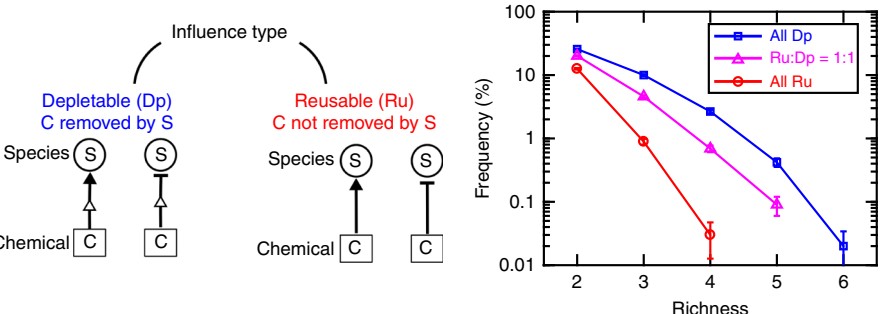

**Fig. 6** Interaction mechanisms impact coexistence. To explore how different interaction mechanisms might influence coexistence, we investigated two categories of chemical interactions: those with depletable mediators (Dp, recipients consume or degrade the chemical mediator) versus those with reusable mediators (Ru, recipients do not affect the mediator concentration). In the extreme cases, communities that used only Dp mediators showed remarkably higher coexistence compared to communities that used only Ru mediators. We also examined enrichment in communities with equal number of the two interaction types, and coexistence in these hybrid communities appeared to be in between the two extremes. In these simulations, links had a binomial network. The initial number of species types $N_c = 20$, the number of possible mediators $N_m = 15$, and the ratio of interaction types $+:- = 50\%:50\%$. Error bars indicate s.d. due to sampling

Our work re-emphasizes the importance of facilitation and self-restraint in community formation and maintenance. Examples of how microbes employ these mechanisms for coexistence are not hard to imagine. Metabolic exchange has been considered as a common way for other-facilitation among microbes[49,58,59], which would fit within the framework of our model. Self-facilitation can take place for example when a species breaks down complex compounds in the environment into consumable products. This can take place by species that produce protease or cellulose; in our model, the mediator will be the product of the breakdown (e.g., amino acids or glucose). The concentrations of such mediators increase in the presence of corresponding species, and those species (and potentially others) will experience a benefit in the presence of such products. Having unique access to these (otherwise inaccessible) resources allows these species to gain a growth rate boost that allows them to persist for coexistence. Self-restraint is also widespread, as the products of metabolism (such as acetate[60] or ethanol[61]) can often become inhibitory when they accumulate in the environment.

We also would like to emphasize that our model suggests that coexistence is affected by the mechanism of interactions among microbes (e.g., interactions mediated through depletable versus reusable chemical mediators). Our earlier work[28] had suggested that pairwise models that do not capture interaction mechanisms fail to properly capture community dynamics. That conclusion is re-iterated here. Models that take the mediators into account appear to address this issue; however, such models are often challenging to experimentally constrain or validate. A rigorous experimental validation requires a well-characterized system with known interactions and chemical mediators. Currently available experimental studies lack this level of detailed characterization. We posit that more experimental examples, along with a better mechanistic understanding of interactions in each case is needed to clarify how and when a mediator-explicit model adequately represents a microbial community.

One of the intriguing observations in our results is the rapid drop in the likelihood of arriving at derived communities with higher richness. Considering that in experimental settings instances of derived communities with several species are not rare[50,62], there is a need to clarify what deviations from our model assumptions may be responsible. Part of the discrepancy might be due to our definition of coexistence; indeed if we define coexistence simply as the presence of species after a shorter period of growth, the likelihood of achieving higher richness increases. Other factors not included in our model may also contribute to coexistence. Temporal or spatial heterogeneity[63,64] could be factors that effectively change the strength of interactions among community members over time. Additionally, in our model shared resources are assumed to be always in abundance; however, if shared resources get depleted, some species that have lower growth rate in abundant resources might be able to persist by growing faster relative to other species when resources are scarce. Environment-dependent fitness tradeoffs have also often been observed. For example, a strain could be more fit than another strain at low metabolite concentrations, but suffer a fitness disadvantage at high metabolite concentrations (e.g., ref. [43]). The additivity assumption in Eq. (1) may also be a factor, as it captures species growth on multiple metabolites[65] but not the situation when more than one mediator is necessary for growth. Previous reports have discussed the exact form that may be more appropriate for representing the combined influence of carbon sources[66,67]. Assessing the generality of those reports for other chemical mediator types require additional investigations.

In assigning the facilitative or inhibitory impact of mediators, we have assumed that no trade-off exists to ensure unbiased sampling of the parameter space from a predetermined distribution. This means that a compound inhibiting or facilitating the growth of one species can at the same time facilitate or inhibit the growth of another species independently. Such a situation can happen in nature; for example in oral biofilms, lactate is inhibitory to some species, while it promotes the growth of other species[4]. However, some structure in the network is expected to exist: for example, byproducts are expected to be lower in energy content compared to input nutrients[30]. Incorporating these structures into the model would be an important step in the future for a more realistic representation of microbial communities.

In modeling microbial coexistence, there are many other aspects that deserve further investigation. Species behavior and physiology may change depending on the environmental cues or intercellular communications[43,68]. There may be an heterogeneity, in terms of microbial phenotypes (either driven by evolution or phenotypic variations), within each population[69]. Another possibility is potential intrinsic structure in the network of the community (e.g., presence of intrinsic modularity), which may cause the coexistence to deviate from predictions of our model based on randomly assigned networks. Non-monotonic interactions[70] and non-additive interactions[34] can also influence the formation and maintenance of microbial communities. These aspects are outside the scope of this report, but can be

**Table 1 Simulation parameters**

| Parameter | Description |
|---|---|
| $N_c = 20$ | Number of cell types in the initial pool |
| $N_m = 10$ | Number of mediators |
| $N_r = 10$ | Number of rounds of transfer (i.e., dilution steps) |
| $N_s = 10,000$ | Number of initial pools of interacting species analyzed |
| $\Sigma S_{init} = 10^4$ | Total initial cell density (ml$^{-1}$) |
| $\alpha = 1$ | Avg. consumption factor per cell (fmol) |
| $\beta = 0.1$ | Avg. production rate per cell (fmol h$^{-1}$) |
| $\rho = 0.2$ | Maximum interaction strength (h$^{-1}$) |
| $\Sigma S_{dil} = 10^{10}$ | Coculture dilution threshold of cell density (ml$^{-1}$) |
| $\Sigma S_{ext} = 0.1$ | Population extinction threshold of cell density (ml$^{-1}$) |
| $dt = 0.01$ | Cell growth update and uptake timescale (h) |
| $K_{sat} = 10^4$ | Influence strength saturation level (fmol ml$^{-1}$) |
| $q_p = 0.2$ | Probability of production link per population |
| $q_c = 0.2$ | Probability of influence link per population |
| $\alpha_{li} \sim U(0.5\alpha, 1.5\alpha)$ | Consumption factors follow a uniform random distribution |
| $\beta_{li} \sim U(0.5\beta, 1.5\beta)$ | Production rates follow a uniform random distribution |
| $K_{l,i} \sim U(0.5K_{sat}, 1.5K_{sat})$ | Influence saturations follow a uniform random distribution |
| $r_i \sim U(0.08, 0.12)$ | Basal net growth rates follow a uniform random distribution (h$^{-1}$) |
| $\sigma_{r0} = st.dev(r_i)$ | Standard deviation of basal net growth rates (h$^{-1}$) |

These parameter values are typically used in our simulations, unless specified explicitly. $U$ is a random number generator with a uniform distribution between its two input arguments

independently examined using the same framework in the future. We hope that this work will be a stepping stone in formulating important features of microbial interactions in community models.

## Methods

**Simulation platform.** Simulations were implemented in Matlab® and run on the Research Services' Linux Cluster at Boston College. For coexistence screens, typically, different sets of assumptions and conditions were simulated (number of samples typically between 3000 and 30,000) and then analyzed to find what aspects impact coexistence. Simulation codes are available at https://github.com/bmomeni/coexistence-via-chemical-interactions. Parameters used in the simulations are defined in Table 1. Parameter values in Table 1 are the ones used in most simulations, unless otherwise specified.

**Formulations for modeling facilitation and inhibition.** We assume that facilitation and inhibition by all chemicals follow a unified form in our model. Based on our characterization data (Fig. 1c and Supplementary Fig. 1), we assume that inhibition by chemicals follows the form

$$r(C_{inh}) = r_0 - r_{inh}\frac{C_{inh}}{K_{inh}} \tag{2}$$

where $r_0$ is the basal net growth rate in the absence of inhibition, $r_{inh}$ represent the strength of inhibition, and $K_{inh}$ determines the dependency on inhibitor concentration. This model is consistent with the simplified view that cells randomly encounter inhibitor molecules that will enter the cell and inhibit their growth with a fixed probability.

We also examined two alternative formulations. Inhibition threshold model (based on Supplementary Fig. 1), in which the effect of inhibition appears only beyond a threshold concentration ($C_{th}$) of the inhibitor (Supplementary Fig. 6B):

$$r(C_{inh}) = \begin{cases} r_0; & C \leq C_{th} \\ r_0 - \frac{r_{inh}}{K_{inh}}(C - C_{th}); & C > C_{th} \end{cases} \tag{3}$$

The "Growth Inhibition" model is based on the formulation in ref. [42]. Briefly, the model incorporates a random chance that an inhibitory molecule enters the cell, and when inside, inhibition slows down growth. As a result, the effect of inhibition is the strongest when the cell grows fast, but further inhibition will not be as strong since cell growth has slowed down (Supplementary Fig. 6B):

$$r(C_{inh}) = \begin{cases} r_0; & C \leq C_{th} \\ r_0 - \frac{r_{inh}}{(1 + K_{inh}/(C - C_{th}))}; & C > C_{th} \end{cases} \tag{4}$$

For facilitation influences, we assume the Monod equation (Fig. 1d),

$$r(C_{fac}) = r_0 + r_{fac}\frac{C_{fac}}{C_{fac} + K_{fac}} \tag{5}$$

where $r_{fac}$ represent the strength of facilitation, and $K_{fac}$ determines the dependency

on facilitator concentration. This model is consistent with the simplified view that cells take up their rate-limiting nutrient according to Michaelis–Menten kinetics and divide when they acquire enough of that nutrient. The more generalized Moser equation (Fig. 1d and Supplementary Fig. 2)

$$r(C_{fac}) = r_0 + r_{fac}\frac{C_{fac}^n}{C_{fac}^n + K_{fac}^n} \tag{6}$$

with exponent $1 < n < 3$ offers a more accurate representation of the dependency on facilitator concentration, but does not have a marked impact on coexistence (Supplementary Fig. 6A).

**Network architecture of initial species pool.** In binomial networks, the presence or absence of c-links and f-links each is determined by a fixed probability. A randomly sampled consumption factor or production rate (see simulation parameters below) is assigned to each consumption or production link, respectively. The basal growth rate values of species in the initial pool of species are picked randomly from a uniform distribution ($r_0 \sim U(0.08, 0.12)$ per hour, except in Supplementary Fig. 9 as noted). The exact value of basal growth rate is inconsequential as all other growth rate values (e.g., the influence of chemicals) and timescales can be scaled accordingly without any loss of generality. For influence strengths, the values are picked randomly from a uniform distribution (see simulation parameters below) when the fraction of positive to negative interactions is 1:1. In cases where either positive or negative interactions are more likely, the absolute values of influence strengths within positive or negative interactions still follow a uniform distribution, but the sign will be positive or negative based on a binomial distribution. The only exception is Supplementary Fig. 7, in which other distributions of influence strengths are used.

**Calculating mean excess richness (MER).** To quantify how much richness is supported in a given setting, we define MER as $\sum_i (i-1)p_i$ where $i$ is the richness (i.e., the number of coexisting species) in the derived community and $p_i$ is the probability of achieving a richness of $i$. This measure quantifies the level of coexistence beyond a single-species domination expected from competitive exclusion. To calculate the confidence intervals for MER, we used bootstrap (using bootci routine in Matlab), typically with 3000 samples with substitution.

**Characterization of chemical facilitation and inhibition.** For facilitation, we examined the growth of *Escherichia coli* K12 MG1655 single gene knockout auxotrophic strains in media supplemented with the corresponding amino acid at different concentrations. For leucine auxotrophy, we replaced LeuB with a chloramphenicol resistance gene and for isoleucine auxotrophy, we replaced IleA with a kanamycin resistance gene. For isoleucine auxotrophs, a BioTek Synergy Mx multimode microplate reader was used to monitor the optical density (OD) cells over 24 h at 5 min intervals. Cultures typically started from an initial OD of 0.001, and were kept shaking in between OD readings. Standard M9 minimal medium (following Cold Spring Harbor Protocols) was used as the basal growth medium in these experiments, and it was supplemented with isoleucine as needed. For leucine auxotrophs, the OD assay above was not sensitive enough to measure the growth

**Table 2 Strains and culture conditions for characterizing the inhibitory effects are listed**

| Strain | Inhibitor | Medium | Temperature (°C) |
|---|---|---|---|
| *E. coli* K12 MG1655 | Acetic acid | M9 | 37 |
| *E. coli* K12 MG1655 | Erythromycin | M9 | 37 |
| *Brevibacillus* M1-5 | Acetic acid | BAAD | 50 |
| *Staphylococcus aureus* SD6 | Acetic acid | 10% THY | 37 |
| *Staphylococcus aureus* SD6 | Erythromycin | 10% THY | 37 |
| *Staphylococcus epidermidis* SD8 | Acetic acid | 10% THY | 37 |

For different species, different inhibitors were introduced in the preferred growth conditions of each species to assess the impact of chemical mediators on cells' growth rate

rate. Instead, we used a fluorescently labeled strain (using DsRed on a plasmid) and used the plate reader to monitor the total fluorescence from the cultures growing when supplemented with different concentrations of leucine. Excitation was set at 560 nm and emission at 588 nm in this assay. We used only the first 3 h of the fluorescence reading to calculate the growth rates to minimize the effect of leucine depletion as cells were growing.

For inhibition, we examined different combinations of bacterial strains and inhibiting compounds, as listed in Table 2.

For these inhibition experiments, we typically streaked them on rich medium (on LB for *E. coli* strains, on PCS for the *Brevibacillus* strain, and on BHI for *Staphylococcus* strains) and isolated a clone. The clone was then grown to exponential phase in the basal media listed in the table, in the absence of inhibitors. Multiple replicate wells on either a 96-well plate or a 384-well plate were inoculated with these exponentially growing cells typically at an initial OD of 0.001, at different concentrations of the corresponding inhibitor. Growth was monitored by recording the OD at 5-min or 10-min intervals using either a BioTek Synergy Mx multi-mode microplate reader, or a BioTek Epoch2 absorbance microplate reader. Plates were incubated while shaking inside the plate reader in between OD readings. Typically 3–6 replicates were used per condition. The wells around the periphery of microplates were found to be more subject to evaporation. We thus filled those with sterile water to reduce the impact of evaporation on other wells and only used the rest of the wells on each plate for our cultures.

**Analyzing the growth rate from experimental OD readings**. To estimate what the growth rate is in each well, we exported the data from Gen5 software that controls microplate readers to a text file, and transferred the data to Matlab for analysis. For each well, we used the wells in time-points 3–10 to estimate the background OD corresponding to that well. The first two time-points were dropped, because we occasionally saw condensation issues before the plate reached the incubation temperature. After subtracting the background, we picked data points for each growth curve that were between OD values of 0.002 and 0.02 to avoid noise at low ODs and saturation at high ODs. A linear function was then fit into the log of OD values using the polyfit function in Matlab. The slope of this line was reported as the growth rate for that well.

**Interpreting coexistence in a continuous growth environment**. For simplicity, we consider a chemostat environment with a constant dilution rate, $\delta$. Additionally, we assume that all mediators (facilitators and inhibitors) have a saturating influence and mediators that are not depleted have concentrations much higher than the saturation concentration, $K_{i,l}$. Therefore, we can simplify the equations

$$\begin{cases} \frac{dS_i}{dt} = \left[ r_{i0} + \sum_l \left( \rho_{il}^+ - \rho_{il}^- \right) \frac{C_l}{C_l + K_{i,l}} \right] S_i - \delta S_i \\ \frac{dC_l}{dt} = \sum_l \left( \beta_{li} S_i - \alpha_{li} \frac{C_l}{C_l + K_{i,l}} S_i \right) - \delta C_l \end{cases} \quad (7)$$

into

$$\begin{cases} \frac{dS_i}{dt} = -\delta S_i + \left[ r_{i0} + \sum_l \left( \rho_{il}^+ - \rho_{il}^- \right) \theta_l \right] S_i \\ \frac{dC_l}{dt} = -\delta C_l + \sum_l \left( \beta_{li} - \alpha_{li} \right) \theta_l S_i \end{cases} \quad (8)$$

where $\theta_l = 1$ for chemical mediators that are accumulated in the environment, and $\theta_l = 0$ for chemicals that are depleted from the environment by cells. If we define $\Theta$ as a matrix with diagonal elements $\theta_l$, we can re-write the equations for population

densities as

$$\frac{d}{dt} \mathbf{S} = (-\delta + \mathbf{r}_0 + \mathbf{P}\Theta)\mathbf{S} \quad (9)$$

Here, $\mathbf{S}$ is the vector containing population densities, $\mathbf{r}_0$ is a diagonal matrix of basal growth rates, and $\mathbf{P}$ is the matrix of influence strengths $\rho_{il}^+ - \rho_{il}^-$. The dilution rate can be adjusted to the growth rate of the sub-community that grows the fastest, representing steady-state coexistence. Thus finding coexistence will be equivalent to finding the subset of species that allow a consistent solution with the largest eigenvalue for the matrix $(\mathbf{r}_0 + \mathbf{P}\Theta)$. This is not a direct problem, because the chemicals that will remain in the environment and their concentrations at steady-state are not known a priori. Nevertheless, certain insights derived from our simulations are consistent with this formulation. For example, both self- and other-facilitation acting on a species are expected to increase the chance of that species being present in the sub-community with the largest eigenvalue. To explain why depletable mediators show higher coexistence, we note that still within the core of species engaged in facilitation, mediator depletion provides an internal feedback to modulate the chemical influence on the species that are most dominant, allowing coexistence of that species with other species. Such a feedback is missing when interaction mediators are reusable.

**Reporting summary**. Further information on research design is available in the Nature Research Reporting Summary linked to this article.

## Data availability
All relevant data is available upon request.

## Code availability
Simulation codes to reproduce the results presented in this paper are available on a GitHub repository at https://github.com/bmomeni/coexistence-via-chemical-interactions.

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

## Acknowledgements

Research in Momeni Lab is supported by the Start-up fund provided by Boston College, and by an Award for Excellence in Biomedical Research from Smith Family Foundation. B.M. also thanks Boston College for Undergraduate Research Fellowships that supported the participation of undergraduate students in this work. The authors would like to thank the Shiaris lab at the University of Massachusetts, Boston for sharing the *E. coli* K12 strain. The authors are grateful to Prof. Shin Haruta from Tokyo Metropolitan University and Prof. Masaharu Ishii at the University of Tokyo for sharing the *Brevibacillus* strain with them. The authors are grateful to Jeff Gore for valuable feedback. The authors also thank Barry Schaudt and Boston College's Research Services for the Linux Cluster used for simulations in this manuscript. B.M. would like to thank Welkin Johnson and Sarah McMenamin for their support in the process of preparing this manuscript.

## Author contributions

W.S. and B.M. conceived the idea. M.L. and B.M. developed the code. K. Chen, D.F., and S. Dedrick performed the inhibition assessment experiments. L.N., C.H., and S. Dyckman performed the facilitation assessment experiments. L.N., I.B., K. Chaung, S.E.M., M.C., and B.M. ran simulations and analyzed the data. L.N., I.B., and B.M. wrote the manuscript. S. Dyckman, S. Dedrick, W.S., and B.M. edited the manuscript.

## Additional information

**Competing interests:** The authors declare no competing interests.

