## [Peer Review File · Nature Communications]

Reviewers' comments:

Reviewer #1 (Remarks to the Author):

In this manuscript the authors present a model to study the effects of chemical-mediated interactions on species coexistence. The model incorporates feedbacks between species growth rates and chemicals that microbes can produce and deplete. By analyzing the different overall microbial interactions that arise from these feedbacks, they show that facilitation and self-restrain promote coexistence. When strong interactions are at play, the topology of the interaction network can determine coexistence, regardless of the specific strengths of the interactions.

In my opinion, the paper is very relevant to microbiologists and also to researchers in other areas, such as ecology. The question of what drives coexistence in microbial communities is still under debate, and recent (and ongoing) works have been focusing on it. Moreover, there is a growing interest on how microbial interactions are mediated through different chemicals in the environment. The model presented in this manuscript is simple but powerful enough to suggest relevant mechanisms promoting coexistence, and it could be easily adapted in the future to accommodate more details on the feedbacks between specific chemicals and microbial species. Overall, I recommend it for publication after a few minor comments are taken into account.

MINOR COMMENTS:

- When describing their model in the last paragraph in page 2, I had the impression that the authors would model a bioreactor scenario. When I read 'continuously growing community' and 'continuous growth simulations' I did not imagine that the authors would actually simulate a discontinuous dilution scenario. I think that the serial dilutions should be mentioned in the introduction.

- In relation to the previous comment, maybe the authors could briefly explain what was the motivation to include this discontinuous death events in the simulations: given that they are not constrained by specific experimental scenarios, a purely continuous scenario seems more general. In order to avoid unbounded growth in such purely continuous model, maybe all microbes should present some degree of self-restrain, but this does not seem unnatural. Also, even if it is just in the discussion section, the authors could mention how removing the discontinuous dilutions could affect their main results.

- In page 6: "The remaining species are considered to coexist within the time-scale of our run". Does this mean that the community could have not reached equilibrium in some cases? It seems to me that, by analyzing a given time window at the end of the simulation, it should not be difficult to say if the community is still changing significantly or not. I suggest to clarify this point.

- The authors use the term 'richness' as a synonym of 'diversity' (number of species in the community). They also use 'enriched' community to describe any community that is present at the end of the simulation. I just wanted to point out that they are using terminology that sounds similar to describe things that are quite different. From the two terms, 'enriched community' is the one for which the choice of words seems less common. I leave to the authors decision whether to change this terminology or not.

- In page 8: "In [51] facilitation is common within the initial pool (as evidenced by coexistence among most pairs studied), and as a result, starting from many pairs and trios coexistence is observed (34 out of 56 pairs and 19 out of 28 trios)". The logic of this sentence seems to be: facilitation is

evidenced by frequent coexistence, and then coexistence is observed. In addition, I don't think it is not true that Friedman et al. found frequent facilitative interactions. In Friedman's paper the outcome of pairs and trios was mostly analyzed in terms of the carrying capacity. In such context facilitation could be detected if the carrying capacity for a given species in coculture is higher than in monoculture, but the authors did not find that this was common. In addition, Friedman et al. also considered a generalized Lotka-Volterra model including mostly competitive (and not cooperative) interactions. When reviewing this paragraph, please also review the next sentence where 'facilitation interactions' are also mentioned.

- Also in the same paragraph in page 8, 'invasion' is used to describe an outcome that may be more commonly known as 'Competitive exclusion' or 'dominance'.

- Page 12, line 3: maybe use 'other parameter values' instead of 'other parameters'.

- Page 16: "if resources become scarce, rate-yield trade-off may allow species with lower intrinsic growth rate to persist". I'm not sure if the authors mean growth rate vs yield trade-offs (which are not necessarily well established). Please clarify and elaborate more about this statement.

- The pH can be a major mediator of interactions in microbial ecosystems, as shown in Ratzke and Gore Plos Biol 2018 for interspecies interactions, and in Ratzke et al Nature Ecol Evol 2018 for self-restrain. It could be nice if the authors include some discussion about pH as an interaction mediator, and whether the pH could be treated or not as any other chemical in this model.

- First equation in page 17 has 2 constants: r_{inh} and K_{inh} . Could they be collapsed into just one? What is the benefit of separating it into two different constants?

- Based on the last sentence in page 17 "in which inhibition becomes less potent when the growth rate decreases", I don't understand the last equation in this page. Don't the two formulations (two last set of equations in page 17) behave the same with respect to the growth rates r_0 and r_{inh} ?

- Refs 50 and 55 point to the same work.

OTHER COMMENTS:

- I found that a very recent review on environmentally mediated interactions by Estrella et al. (TRENDS Ecol Evol 2018) could be relevant to the authors. However, I leave to the author's decision whether it should be cited or not.

Reviewer #2 (Remarks to the Author):

In their manuscript "Microbial coexistence through chemical-mediated interactions" Neihaus and colleagues use mathematical models to investigate the role of interspecies interactions – particularly those mediated by production and consumption of chemicals – in maintaining community diversity.

How to maintain a diverse microbial community is an important question in microbial ecology, and in ecological theory more broadly. Much of the criticism of previous work here has centered on the use of very simple models for microbial interactions; Niehus and colleagues' extension to more mechanistic models thus addresses an important gap in the field.

Broadly I found this manuscript well-written and well-conceived, however, I had one primary criticism that I believe needs addressing at least verbally, but ideally mathematically prior to publication. Namely, in this model microbial facilitation is additive, potentially unlimited, and (of lesser importance) facultative. Specifically, organisms bear no cost of enhancing one another's growth and can do so in an effectively infinite manner. As a consequence, somewhat unsurprisingly communities will have the best chance of maintaining diversity if all species help one another as much as possible. In reality, biology is characterized by trade-offs. Microbes often apportion their investment in digesting different nutrient sources depending on availability and preference (e.g. Bacteroides changing expression of different polysaccharide utilization loci) and cannot support an infinite repertoire of different enzymes at any one time. Conversely, while cross-feeding via waste products can be to a certain extent cost free, rare examples of costly facilitation have been observed in microbial communities. How would implementing these restrictions influence the results seen? I believe this question at least merits further discussion, but ideally could also be addressed mathematically.

Smaller comments:

Stability here has been defined as the ability to remove a link and still maintain species. How do other common stability measures compare – for example, linear asymptotic stability?

I appreciated the work demonstrating that the exact formulation of inhibition / facilitation didn't qualitatively affect the outcomes, I would consider bringing these figures into the main text at least in part as it's one of the first concerns I had.

Figure 4 is quite unclear, for example at first sight it looks like you have two different values for initial fraction of facilitation, can you present these data in a different way? Also, there seems to be a bimodal distribution of final fraction of self-inhibition (at 1 and 0.5) – what is driving this?

Page 12 final paragraph – competition “hitchhiking” implies competition is able to exist as a function of the facilitation being here. I would remove this sentence or explain / prove this point further.

Fig 5 – N=106 for negative influences on others vs N=9511 for positive on others – what is the reason for this difference given you're starting from pools of 50:50?

I appreciated the section dealing with depletable resources, arguably this is the far more biologically relevant case and should be the primary case examined rather than an afterthought. I'm curious however, are these depletable resources introducing competition between species (over the resources), or simply acting as an additional cap on the level of facilitation possible. Either way I think this section is important and could do with further discussion.

Related to this, Eqn1 is the formulation for dC/dt in the main text correct? Which scenario for depletion / reusability is it referring to? Please make this clearer.

Reviewer #3 (Remarks to the Author):

The manuscript “Microbial coexistence through chemical-mediated interactions” describes a (for the most part) computational study of various factors affecting the diversity of microbial communities growing in a cyclically-diluted turbidostat. One of the new factors considered in this study is that, in

addition to cross-feeding, species could also reduce each other's growth by secreting either reusable or consumable molecular "mediators". The results are interesting and the study is at the level suitable for publication in Nature Communications. There are a few omissions/confusing statements I would like authors to correct first:

1) On page 2 and then subsequently on page 6 authors emphasize the fundamental differences between their model and Consumer-Resource Models (CRMs) involving cross-feeding between species. I think these differences are for the most part illusory. Indeed, species in the turbidostat-like setup used in this study compete with each for the highest growth rate in the interval between the dilutions. This is no different from species in a (for example) batch-fed chemostat competing for the most efficient use of rate-limiting resources. In both case the surviving species are selected based on the appropriate average of their growth function $g_i(C_1, C_2 \dots C_K)$ with all co-existing species having this average equal to each other. Every species that went extinct during the transient period did so because its average g_i turned out to be smaller than that of the survivors. I would like authors to properly explain the similarities and differences between their system and various types of chemostats and turbidostats (see page 20 where for the sake of simplicity they replace their system with a continuously diluted chemostat). Authors should also remove the mention of how their model is fundamentally different from CRMs (it is not).

2) The appropriate measure of the strength of positive and negative interactions between species is not ρ/r_0 used e.g. in Fig. 3 and around, but $\rho/\text{Std}_{\text{species}}(r_0)$. Indeed, species use cross-feeding and negative interactions to eliminate the differences in their intrinsic growth rates quantified by $\text{Std}_{\text{species}}(r_0)$. It took me some sleuthing to finally find that in their simulations authors use r_0 drawn from $U(0.08 \text{ and } 0.12)$. Its standard deviation is only 0.0115 (compared to the mean of 0.1). Hence the relevant "relative strength" $\rho/\text{Std}(r_0)$ would be 10 times larger than ρ/r_0 . My hunch is that the mean value of r_0 (for a given value of $\text{Std}(r_0)$) would have minimal effect (if any at all) on community's diversity. This need to be numerically verified/disproved. On a somewhat related note, whenever authors study the effect of some parameter on diversity, they quantify it as a bunch of plots with PDFs of the number of surviving species. It would be nice also to have a separate panel showing $\langle n_{\text{surv}} \rangle - 1$ plotted vs. the parameter in question. This would help readers to quantify the reported trends. Note that I subtracted 1 to be able to use (whenever appropriate) the (double) logarithmic axes in these plots. Indeed, in the turbidostat setup the minimal diversity of 1 is guaranteed. Hence it should be subtracted to show the "mean excess diversity". This comment applies to most figures in this study. I want authors to (whenever appropriate and illuminating) add an extra panel to their figures showing the trends of the mean excess diversity.

3) In their systematic effort to reveal all the determinants of community's diversity, authors overlooked one important factor – the mean rate of dilution δ (for their system δ given by something like $\log(\text{Th}_{\text{dil}}/\text{CSD})/(\text{average time between consecutive dilutions})$). According to the results of

<https://journals.aps.org/prl/abstract/10.1103/PhysRevLett.120.158102> subsequently reproduced in Ref. 43 and cited in the manuscript, higher dilution rates in general tend to reduce the diversity. This could be verified (or disproved) for the system in this study.

4) The exact rules of when and how their turbidostat is being diluted are not properly explained in the manuscript. My understanding is that the system is cyclically diluted (and not periodically, as authors state in the caption to Fig. 2). The dilution happens whenever the overall cell density reaches a predetermined threshold $\text{Th}_{\text{dil}} = 10^{10}$ cells/mL (which I had to fish out from the Methods section). The system is then diluted down not by a constant factor but to another predetermined cell density $\text{CSD} = 10^4$ cells/mL. This density is referred to as the "initial cell density" on page 19. Is it just the density at the start of the simulation or it reappears in the beginning of every cycle? Authors need to clear up this confusion. It would be also most illuminating if authors were to move an illustration of the cyclic dynamic shown in Fig. S3C to the main text of the manuscript.

Some minor typos: on page 3 "growth promotion influences" -> "growth-promoting influences".

Response to Reviewers' comments:

We would like to thank the editor and reviewers for their helpful and constructive feedback. We have revised the manuscript accordingly. Below please find the list of changes made. In the attached revised document, all the changes are marked in a **blue font** to make it easier for the reviewers to examine the modifications.

Reviewer #1:

In this manuscript the authors present a model to study the effects of chemical-mediated interactions on species coexistence. The model incorporates feedbacks between species growth rates and chemicals that microbes can produce and deplete. By analyzing the different overall microbial interactions that arise from these feedbacks, they show that facilitation and self-restrain promote coexistence. When strong interactions are at play, the topology of the interaction network can determine coexistence, regardless of the specific strengths of the interactions.

In my opinion, the paper is very relevant to microbiologists and also to researchers in other areas, such as ecology. The question of what drives coexistence in microbial communities is still under debate, and recent (and ongoing) works have been focusing on it. Moreover, there is a growing interest on how microbial interactions are mediated through different chemicals in the environment. The model presented in this manuscript is simple but powerful enough to suggest relevant mechanisms promoting coexistence, and it could be easily adapted in the future to accommodate more details on the feedbacks between specific chemicals and microbial species. Overall, I recommend it for publication after a few minor comments are taken into account.

> Thank you very much for your support.

MINOR COMMENTS:

- When describing their model in the last paragraph in page 2, I had the impression that the authors would model a bioreactor scenario. When I read 'continuously growing community' and 'continuous growth simulations' I did not imagine that the authors would actually simulate a discontinuous dilution scenario. I think that the serial dilutions should be mentioned in the introduction.

> We have revised the manuscript and defined the term "continuous growth" in the introduction (Page 2):

"The term 'continuous growth' emphasizes that external resources are replenished at each passage to be in excess and species growth rates are modulated by species interactions. Continuous growth can be found in environments such as turbidostats, some industrial bioreactors, or possibly human gut in which resources are continuously or cyclically supplied³⁹. This model can be considered as a special type of consumer-resource model^{29,30,40} in which chemical mediators generated by species are modeled, but external resources are not modeled since they are supplied in excess."

We have also included an additional supplementary figure (Supplementary Fig 8) to directly address the effect of dilution magnitude on coexistence. We observe that the dilution scheme does not considerably affect the coexistence outcome.

Supplementary Fig 8. Coexistence is not sensitive to the dilution scheme. The likelihood of coexistence is only modestly affected by the dilution factor (i.e. the ratio of total population density before and after dilution). (A-B) As the range of population growth around a central population density expands (insets) from a 2-fold range (from 5×10^5 to 10^6 cells/ml) to a 5000-fold range (from 10^4 to 5×10^7 cells/ml), coexistence decreases, but gradually. (C) Mean excess richness measure (defined as average richness in derived communities, beyond competitive exclusion) confirms a gradual drop in coexistence when the dilution factor increases. Error bars show bootstrap estimates of 95% confidence intervals for the mean values. In these simulations, $\rho/\sigma_{r0} = 20$, the number of initial species types $N_c=20$, the number of mediators $N_m=10$, and the number of initial pools examined $N_s=5000$. Using a representative example with $N_c=10$ and $N_m=5$, we observe that (D) increasing the dilution step from 10x (top) to 100x (bottom) does not considerably change the population dynamics in the community, when the central population density does not change much. Further increasing the dilution step to 10^4 x (E, top) leads to some qualitative changes, such as persistence of the cyan population beyond 200 hours. This is because the inhibition of the cyan population is weaker in (E), in which there is lower inhibitor concentration (and fewer inhibitor producers) at the beginning of each round. When we shift the densities to the center of the range in (E, top), dynamics become qualitatively similar (E, bottom), even with a gentle 2-fold dilution. Total central cell density (cells/ml) and fold-dilution are marked in the inset of each panel.

- In relation to the previous comment, maybe the authors could briefly explain what was the motivation to include this discontinuous death events in the simulations: given that they are not constrained by specific experimental scenarios, a purely continuous scenario seems more general. In order to avoid unbounded growth in such purely continuous model, maybe all microbes should present some degree of self-restrain, but this does not seem unnatural. Also, even if it is just in the discussion section, the authors could mention how removing the discontinuous dilutions could affect their main results.

> Our main motivation was to stay close to a typical experimental setup in which the community goes through rounds of growth and dilution. We have revised the text to clarify this motivation (Page 6):

“To obtain communities that exhibit species coexistence, we simulate cycles of growth and dilution to emulate a typical experimental setting^{39,47-49} called “enrichment”^{50,51}. We call the resulting communities “derived communities”. Since shared resources are replenished cyclically to be in excess, cells are not limited by shared resources, but instead grow at a rate dictated by (Eq 1).”

The added supplementary figure (Supplementary Fig 8) addresses the limit when the dilution steps are very gentle. The observation is that larger dilution steps only modestly reduce coexistence (Page 9):

“We also find that the dilution scheme has only a modest influence on coexistence, with more strict dilutions leading to less coexistence (Supplementary Fig 8), consistent with previous reports⁵⁴.”

- In page 6: “The remaining species are considered to coexist within the time-scale of our run”. Does this mean that the community could have not reached equilibrium in some cases? It seems to me that, by analyzing a given time window at the end of the simulation, it should not be difficult to say if the community is still changing significantly or not. I suggest to clarify this point.

> We have operationally defined coexistence within a fixed time-scale to make it easier to directly compare the results with experiments.

The definition of coexistence is explicitly included in the revised version (Pages 6-7):

“We define “coexistence” based on species that persist in the process of enrichment. Choosing the definition of coexistence faces a tradeoff. If all species that are present within a time frame are considered to coexist, we will inevitably include species that will eventually go extinct beyond the time frame. On the other hand, if we include only instances that we can verify to truly show coexistence over a very long term, then we will deviate from experimental feasibility as such a verification is unlikely implemented. Thus, we have chosen a balance between these two extremes, adopting an experimentalist’s point-of-view (for example, as in Refs. 49,52). Specifically, we have defined coexistence operationally as persistence of species in the community after a given amount of community growth (here, 200 generations). If the population size of a species drops below one cell, then the species is considered extinct and removed from the rest of the simulation. If the population size of a species drops by more than 10% in the final 20 generations of community growth, we also remove it from coexistent communities, because it will presumably slowly go extinct. This definition of coexistence is consistent with “long-term stability” in the sense that in most cases, if we had extended growth time without introducing perturbations, species would continue to coexist (Supplementary Fig 4). Our definition of coexistence is also consistent with “asymptotic stability” in the sense that in majority of cases, if we perturb species frequency away from the steady state value, species frequency will return to the original steady state value (Supplementary Fig 4).”

We have added an additional supplementary figure (Supplementary Fig 4) to clarify our definition of coexistence and its relation to stability.

Supplementary Fig 4. Coexistence is closely related to “long-term stability” and “asymptotic stability”. We examined long-term stability by extending the range of growth without introducing any perturbations. Here each round corresponds to 20 generations of community growth. In a representative example (A), enrichment leads to a derived community that shows coexistence of three species (B). In most cases ($12\% \pm 7\%$ s.d. among 1650 cases examined) communities that showed coexistence also exhibited long-term stability (G, left), similar to the example in (C). We also examined the asymptotic stability of community compositions, by changing the fraction of one of the member species by 3x and asking whether the community composition would recover to the initial fractions afterwards (D-F). Recovery was assessed based on whether after 200 additional generations of community growth, the difference between fractions of species in the unperturbed versus perturbed community was less than 10%. Among communities that exhibit long-term stability, a majority of communities also demonstrated asymptotic stability ($93\% \pm 5\%$ s.d. among 1446 cases examined).

- The authors use the term 'richness' as a synonym of 'diversity' (number of species in the community). They also use 'enriched' community to describe any community that is present at the end of the simulation. I just wanted to point out that they are using terminology that sounds similar to describe things that are quite different. From the two terms, 'enriched community' is the one for which the choice of words seems less common. I leave to the authors' decision whether to change this terminology or not.

> To avoid confusion, the communities obtained through enrichment are now called "derived" communities.

- In page 8: "In [51] facilitation is common within the initial pool (as evidenced by coexistence among most pairs studied), and as a result, starting from many pairs and trios coexistence is observed (34 out of 56 pairs and 19 out of 28 trios)". The logic of this sentence seems to be: facilitation is evidenced by frequent coexistence, and then coexistence is observed. In addition, I don't think it is not true that Friedman et al. found frequent facilitative interactions. In Friedman's paper the outcome of pairs and trios was mostly analyzed in terms of the carrying capacity. In such context facilitation could be detected if the carrying capacity for a given species in coculture is higher than in monoculture, but the authors did not find that this was common. In addition, Friedman et al. also considered a generalized Lotka-Volterra model including mostly competitive (and not cooperative) interactions. When reviewing this paragraph, please also review the next sentence where 'facilitation interactions' are also mentioned.

> Our previous finding emphasized the critical role of facilitation on coexistence [62]. The argument is that if facilitation is prevalent among a group of species, the expectation is that coexistence will be more likely among pairs (and also trios) picked from this group. This is consistent with Friedman et al's and Higgins et al's data, even though the effect of species on the growth rate of other species is not directly measured in their experiments. We have revised the writing accordingly (Page 9):

"In Ref. 52, coexistence was observed among most pairs studied (34 out of 56 pairs), suggesting that many species pairs might be engaged in facilitation⁵⁵. Assuming this is the case, we would expect that many trios would also show coexistence, which is consistent with the observed results (19 out of 28 trios). In Ref. 48, there are fewer instances of pairwise coexistence (19 out of 190 pairs) and more instances of bistability (15 out of 190 pairs), suggesting fewer facilitation and more inhibition among these species, compared to Ref. 52. Interestingly, starting from a pool of all twenty species in Ref. 48, the only trio that showed coexistence had species that all coexisted in pairs as well. This is consistent with the speculation that these three species facilitate each other's growth."

- Also in the same paragraph in page 8, 'invasion' is used to describe an outcome that may be more commonly known as 'Competitive exclusion' or 'dominance'.

> We have used the term 'competitive exclusion' in the revised manuscript.

- Page 12, line 3: maybe use 'other parameter values' instead of 'other parameters'.

> The term is corrected per your suggestion.

- Page 16: "if resources become scarce, rate-yield trade-off may allow species with lower intrinsic

growth rate to persist". I'm not sure if the authors mean growth rate vs yield trade-offs (which are not necessarily well established). Please clarify and elaborate more about this statement.

> Even if not a universal trend, we and others have observed a trade-off that the relative fitness of species can depend on resources abundance. For instance, among lysine-requiring yeast cells, we have observed that mutants with higher affinity for lysine have a growth rate advantage over wild-type at low lysine concentrations, but the trend reverses at high lysine concentrations (Hart et al, 2018). We have revised the manuscript for clarification (Page 18):

“Environment-dependent fitness tradeoffs have often been observed. For example, a strain could be more fit than another strain at low metabolite concentrations, but suffer a fitness disadvantage at high metabolite concentrations (e.g. Ref. 43).”

- The pH can be a major mediator of interactions in microbial ecosystems, as shown in Ratzke and Gore Plos Biol 2018 for interspecies interactions, and in Ratzke et al Nature Ecol Evol 2018 for self-restrain. It could be nice if the authors include some discussion about pH as an interaction mediator, and whether the pH could be treated or not as any other chemical in this model.

> Thank you for the suggestion. pH (treated as $[H^+]$) can be modeled similar to other chemical mediators in this model, as long as its influence on cells is abstracted into a change in growth rate. We have added the following discussion to the manuscript (Page 6):

“This modeling platform is fairly general and can capture a variety of inhibitory and facilitative chemical interactions⁴⁴. Such interactions can include for example the effect of pH (modeled as the concentration of H^+), which is known to impact community structure^{45,46}.”

- First equation in page 17 has 2 constants: r_{inh} and K_{inh} . Could they be collapsed into just one? What is the benefit of separating it into two different constants?

> We have clarified this (Page 4):

“Even though ρ_{il}^- and $K_{i,l}$ can be collapsed into a single term, we have chosen to use the current form so that we can directly compare ρ_{il}^- with ρ_{il}^+ .”

- Based on the last sentence in page 17 “in which inhibition becomes less potent when the growth rate decreases”, I don't understand the last equation in this page. Don't the two formulations (two last set of equations in page 17) behave the same with respect to the growth rates r_0 and r_{inh} ?

> We have clarified this (Page 20):

“The ‘Growth Inhibition’ model is based on the formulation in Ref. 42. Briefly, the model incorporates a random chance that an inhibitory molecule enters the cell, and when inside, inhibition slows down growth. As a result, the effect of inhibition is the strongest when the cell grows fast, but further inhibition will not be as strongly since cell growth has slowed down. (Supplementary Fig 6B).”

- Refs 50 and 55 point to the same work.

> The issue is fixed in the revised version.

OTHER COMMENTS:

- I found that a very recent review on environmentally mediated interactions by Estrella et al. (TRENDS Ecol Evol 2018) could be relevant to the authors. However, I leave to the author's decision whether it should be cited or not.

> Thank you for the suggestions. We have revised the manuscript and included this reference.

Reviewer #2 (Remarks to the Author):

In their manuscript “Microbial coexistence through chemical-mediated interactions” Neihaus and colleagues use mathematical models to investigate the role of interspecies interactions – particularly those mediated by production and consumption of chemicals – in maintaining community diversity.

How to maintain a diverse microbial community is an important question in microbial ecology, and in ecological theory more broadly. Much of the criticism of previous work here has centered on the use of very simple models for microbial interactions; Neihaus and colleagues’ extension to more mechanistic models thus addresses an important gap in the field.

> Thank you for your support.

Broadly I found this manuscript well-written and well-conceived, however, I had one primary criticism that I believe needs addressing at least verbally, but ideally mathematically prior to publication. Namely, in this model microbial facilitation is additive, potentially unlimited, and (of lesser importance) facultative. Specifically, organisms bear no cost of enhancing one another’s growth and can do so in an effectively infinite manner. As a consequence, somewhat unsurprisingly communities will have the best chance of maintaining diversity if all species help one another as much as possible. In reality, biology is characterized by trade-offs. Microbes often apportion their investment in digesting different nutrient sources depending on availability and preference (e.g. Bacteroides changing expression of different polysaccharide utilization loci) and cannot support an infinite repertoire of different enzymes at any one time. Conversely, while cross-feeding via waste products can be to a certain extent cost free, rare examples of costly facilitation have been observed in microbial communities. How would implementing these restrictions influence the results seen? I believe this question at least merits further discussion, but ideally could also be addressed mathematically.

> Thank you for the thoughtful comment. We agree with the reviewer that incorporating some of the trade-offs would be an important step towards making the model more realistic. However, currently there is not enough data to properly constrain the model with a realistic trade-off. Even though some structure is intuitive (e.g. byproducts are expected to be lower in energy content compared to input nutrients), other trends may be less obvious especially when a compound that inhibits the growth of one species can at the same time inhibit or facilitate another species’ growth to independently different extents. Such a situation can happen in nature; for example, lactate is inhibitory to some species, while it promotes the growth of other species in oral biofilms. In the current form, we have intentionally removed any such structure to allow an unbiased sampling of the parameter space. We are considering incorporating the trade-offs into the model (for example, by combining our model with (Marsland et al 2018) which has the structure for resources but does not include strict inhibition). We have added the following paragraph to acknowledge and emphasize the need for incorporating the trade-off in the future (Page 19):

“In assigning the facilitative or inhibitory impact of mediators, we have assumed that no trade-off exists to ensure unbiased sampling of the parameter space from a predetermined distribution. This means that a compound inhibiting or facilitating the growth of one species can at the same time facilitate or inhibit the growth of another species independently. Such a situation can happen in nature; for example in oral biofilms, lactate is inhibitory to some species, while it promotes the growth of other species⁴. However in a realistic community, some structure in the network is expected to exist: for example, byproducts are expected to be lower in energy content compared to input nutrients³⁰. Incorporating these structures into the model would be an important step in the future for a more realistic representation of microbial communities.”

Smaller comments:

Stability here has been defined as the ability to remove a link and still maintain species. How do other common stability measures compare – for example, linear asymptotic stability?

> We have observed that communities in which species stably coexist in our model will often exhibit asymptotic stability, as assessed by return to the original composition after an imposed perturbation in community composition. We have added a supplementary figure to explicitly address this point (Please see the response to the third comment by Reviewer #1, and the associated Supplementary Fig 4).

I appreciated the work demonstrating that the exact formulation of inhibition / facilitation didn't qualitatively affect the outcomes, I would consider bringing these figures into the main text at least in part as it's one of the first concerns I had.

> To address your concern, in the revised version, we emphasize early on that coexistence is affected by the strength of chemical-species influences, but not the details of how they change with concentrations (Page 6):

“In what follows, we will use this model to examine how chemical interactions among microbes may allow different species to coexist. In our analysis of coexistence, we will rely on the experimentally-motivated model formulation in (Eq 1). Nevertheless, we will show that our findings depend on the sign and strength of interactions (Fig 3), but not on the details of this formulation (Supplementary Fig 6).”

Figure 4 is quite unclear, for example at first sight it looks like you have two different values for initial fraction of facilitation, can you present these data in a different way? Also, there seems to be a bimodal distribution of final fraction of self-inhibition (at 1 and 0.5) – what is driving this?

> We have now marked the regions in which facilitation or inhibition is favored on each plot to clarify the purpose of the graph. Additionally, we presented the summary of Fig 4 as a simplified Pi chart.

The bimodal distribution of final fraction of self-inhibition is caused by the small number of inhibitory influences remained in the derived communities. With these small numbers, the fractions appear to be quantized (1 out of 1, 1 out of 2, 2 out of 2, 1 out of 3, etc.). A description is added to the caption of the figure to make this point clear (Caption of Fig 4; Page 13):

Fig 4. Facilitation and self-restraint are favored in enrichment

We examined how the fraction of interaction types changed from the initial pool to the final derived community. Each dot here represents an enrichment process going from twenty initial species to a final community with at least two coexistent species. We examined three conditions with the initial ratio of facilitation to inhibition influences being 20:80 (red), 50:50 (black), or 80:20 (blue). (A) Facilitation was highly favored (i.e. its frequency increased from the initial pool to the derived communities), and 90% of derived communities contained only facilitation. (B) In final communities that contained inhibition, self-restraint was prevalent (82% in this category contained only self-restraint), even though the fraction of self-restraint was only $20 \pm 11\%$ (s.d.) in the initial pool. Points above the diagonal line represent that facilitation and self-restraint are favored in (A) and (B), respectively. Fractions in derived communities appear “quantized” (especially in B) due to the small numbers of influences remaining (1 out of 1, 1 out of 2, 2 out of 2, 1 out of 3, etc.). The data points in (A) and (B) are jittered by 5% to reduce overlap for visualization purposes. (C) We examined the break-down of different categories of influences. Facilitation was favored in 96% of communities during enrichment, including in all communities that lacked self-restraint. Among the 4% in which facilitation was disfavored, in 95% of cases self-restraint was favored during enrichment. This suggests that facilitation is the main driver of coexistence, with self-restraint being a secondary means of achieving coexistence. In these simulations, the number of initial pools examined $N_s=30000$, the initial number of species types $N_c=20$, and the number of possible mediators $N_m=15$. All mediators in these simulations are depletable; the same trends hold when mediators are all reusable.

Page 12 final paragraph – competition “hitchhiking” implies competition is able to exist as a function of the facilitation being here. I would remove this sentence or explain / prove this point further.

> We have added the following sentences to the manuscript to make this point clearer (Page 14):

“In other words, a member species that is engaged in several facilitation interactions with other species may be able to tolerate one or more inhibitory influences. If such inhibitory influences are present in the initial pool, they might not prevent coexistence and thus persist to the derived communities.”

Also, we have added more description on

“In each simulation, we initially put together several species at equal proportions with a random network of interactions. These communities grow (following Eq 1) from a set total initial cell density (ΣS_{init}) up to a pre-determined threshold cell density for dilution (ΣS_{dil}), upon which the culture is diluted back to the initial cell density (Fig 2A, top). Therefore, species with larger growth within each round will be over-represented in the next round. This can be considered as competition for “space” in the inoculum for the next round, leading to the coexistence of species that overall (because of their basal growth rate and interactions exerted by other species) grow the fastest (Fig 2A, bottom). In this setting, each species can grow on its own in the supplied shared resources in the absence of chemical mediated interactions.”

Fig 5 – N=106 for negative influences on others vs N=9511 for positive on others – what is the reason for this difference given you’re starting from pools of 50:50?

> The influence links examined are picked from derived communities, not the original pool of 50:50. As a result, there are fewer instances of negative influences compared to positive ones. This point is now explicitly mentioned in the revised version (Caption of Fig 5; Page 15):

“Since examined influences are chosen from derived communities that exhibited coexistence, more facilitative influences are present (e.g. Fig 4) and thus tested (n=13131) than inhibitory influences (n=775).”

I appreciated the section dealing with depletable resources, arguably this is the far more biologically relevant case and should be the primary case examined rather than an afterthought. I’m curious however, are these depletable resources introducing competition between species (over the resources), or simply acting as an additional cap on the level of facilitation possible. Either way I think this section is important and could do with further discussion. Related to this, Eqn1 is the formulation for dC/dt in the main text correct? Which scenario for depletion / reusability is it referring to? Please make this clearer.

> Consumption of mediators that facilitate the growth of other species can be conceptually treated as competition. Similarly, consumption of mediators that inhibit the growth of other species effectively provides a benefit to them. Because the effect can be positive or negative, we have not discussed consumption in the context of competition. Consumption is an integral part of our model (Eq 1), and the case with only reusable mediators is a special case in which all consumption rates are set to zero.

Regarding dC/dt , the equation represents the most general case, including the consumption of mediators by

species. In the special case of reusable mediators, the consumption rates, α , are set to zero. This point is now emphasized for more clarity in the revised version (Page 5):

“The consumption/degradation of mediators is included in this formulation through consumption factors, α_{li} . Note that consumption of mediators that facilitate the growth of other species can be conceptually treated as competition. Similarly, consumption of mediators that inhibit the growth of other species effectively provides a benefit to them. If a mediator is consumed/degraded by a cell, we call it a *depletable* mediator. In the special case of *reusable* mediator, cells are affected by the mediator without considerably consuming or degrading it (e.g. in response to a signaling molecule), and we set $\alpha_{li} = 0$.”

Reviewer #3 (Remarks to the Author):

The manuscript “Microbial coexistence through chemical-mediated interactions” describes a (for the most part) computational study of various factors affecting the diversity of microbial communities growing in a cyclically-diluted turbidostat. One of the new factors considered in this study is that, in addition to cross-feeding, species could also reduce each other’s growth by secreting either reusable or consumable molecular “mediators”. The results are interesting and the study is at the level suitable for publication in Nature Communications. There are a few omissions/confusing statements I would like authors to correct first:

1) On page 2 and then subsequently on page 6 authors emphasize the fundamental differences between their model and Consumer-Resource Models (CRMs) involving cross-feeding between species. I think these differences are for the most part illusory. Indeed, species in the turbidostat-like setup used in this study compete with each for the highest growth rate in the interval between the dilutions. This is no different from species in a (for example) batch-fed chemostat competing for the most efficient use of rate-limiting resources. In both case the surviving species are selected based on the appropriate average of their growth function $g_i(C_1, C_2 \dots C_K)$ with all co-existing species having this average equal to each other. Every species that went extinct during the transient period did so because its average g_i turned out to be smaller than that of the survivors. I would like authors to properly explain the similarities and differences between their system and various types of chemostats and turbidostats (see page 20 where for the sake of simplicity they replace their system with a continuously diluted chemostat). Authors should also remove the mention of how their model is fundamentally different from CRMs (it is not).

> We have clarified our manuscript (Page 3):

“This model can be considered as a special type of consumer-resource model^{29,30,40} in which chemical mediators generated by species are modeled, but external resources are not modeled since they are supplied in excess.”

Also, on Page 6:

“In each simulation, we initially put together several species at equal proportions with a random network of interactions. These communities grow (following Eq 1) from a set total initial cell density (ΣS_{init}) up to a pre-determined threshold cell density for dilution (ΣS_{dil}), upon which the culture is diluted back to the initial cell density (Fig 2A, top). Therefore, species with larger growth within each round will be over-represented in the next round. This can be considered as competition for “space” in the inoculum for the next round, leading to the coexistence of species that overall (because of their basal growth rate and interactions exerted by other species) grow the fastest (Fig 2A, bottom). In this setting, each species can grow on its own in the supplied shared resources in the absence of chemical mediated interactions.”

2) The appropriate measure of the strength of positive and negative interactions between species is not ρ/r_0 used e.g. in Fig.3 and around, but $\rho/Std_species(r_0)$. Indeed, species use cross-feeding and negative interactions to eliminate the differences in their intrinsic growth rates quantified by $Std_species(r_0)$. It took me some sleuthing to finally find that in their simulations authors use ρ/r_0 drawn from $U(0.08 \text{ and } 0.12)$. Its standard deviation is only 0.0115 (compared to the mean of 0.1). Hence the relevant “relative strength” $\rho/Std(r_0)$ would be 10 times larger than ρ/r_0 . My hunch is that the mean value of r_0 (for a given value of $Std(r_0)$) would have minimal effect (if any at all) on community’s diversity. This need to be numerically verified/disproved. On a somewhat related note, whenever authors study the effect of some parameter on diversity, they quantify it as a bunch of

plots with PDFs of the number of surviving species. It would be nice also to have a separate panel showing $\langle n_{\text{surv}} \rangle - 1$ plotted vs. the parameter in question. This would help readers to quantify the reported trends. Note that I subtracted 1 to be able to use (whenever appropriate) the (double) logarithmic axes in these plots. Indeed, in the turbidostat setup the minimal diversity of 1 is guaranteed. Hence it should be subtracted to show the “mean excess diversity”. This comment applies to most figures in this study. I want authors to (whenever appropriate and illuminating) add an extra panel to their figures showing the trends of the mean excess diversity.

> Thank you very much for the suggestion. You are correct that $\text{Std}(r_0)$ is the more relevant factor for normalization. We have revised the figures accordingly, and throughout the revised version, we report ρ/σ_{r_0} .

We have added a supplementary figure (Supplementary Fig 9) to address your concern regarding the effect of r_0 versus $\text{Std}(r_0)$. Coexistence appears to be more sensitive to $\text{Std}(r_0)$, as predicted (Page 10):

“We use ρ/σ_{r_0} as a measure of the strength of interactions: through interactions mediated by chemicals (with average strength ρ), species achieve coexistence by compensating the difference in their basal growth rates, quantified as the standard deviation of basal growth rates σ_{r_0} .”

“When the interactions are strong, coexistence appears to be insensitive to the mean basal growth rates of species in the initial pool, but dependent on the standard deviation of basal growth rates (Supplementary Fig 9). This is consistent with the intuition that interactions have to compensate for the differences in basal growth rates and a larger standard deviation makes coexistence less likely (with other parameters fixed). To quantify the level of coexistence in each case, we have defined “mean excess richness” (MER) as the average richness beyond single-species dominance predicted by competitive exclusion principle, in derived communities across all sampled initial pools (Methods). MER trends (Fig 3C) confirm the saturating trends observed in Fig 3A-B.”

Supplementary Fig 9. Coexistence is sensitive to the standard deviation of the basal growth rates in the initial pool. (A) As the standard deviation of basal growth rates in the initial pool (σ_{r_0}) increases, the likelihood of coexistence decreases. The mean of basal growth rates (r_0) is kept constant in these cases. (B) The likelihood of coexistence appears to be fairly insensitive to the mean basal growth rate in the initial pool, when σ_{r_0} is kept fixed. Error bars in (A) and (B) are standard deviations based on counting uncertainty. (C-D) Mean excess richness values (average number of coexistent species beyond competitive exclusion) agree with qualitative trends in (A) and (B). Error bars in (C) and (D) show bootstrap estimates of 95% confidence intervals for the mean values.

To report the trends, we now have added the suggested measure to the manuscript. Since we have used richness to describe the diversity throughout the paper, we have used the term ‘mean excess richness,’ even though the definition matches your suggestion (Page 22):

“Calculating mean excess richness (MER): To quantify how much richness is supported in a given setting, we define MER as $\sum_i (i - 1)p_i$ where i is the richness (i.e. the number of coexisting species) in the derived community and p_i is the probability of achieving a richness of i . This measure quantifies the level of coexistence beyond a single-species domination expected from competitive exclusion. To calculate the confidence intervals for MER, we used bootstrap (using `bootci` routine in Matlab), typically with 3000 samples with substitution.”

Figure 3, Supplementary Fig 8, and Supplementary Fig 9 in the revised version include MER to quantify the level of richness in derived communities in an ensemble of cases tested.

3) In their systematic effort to reveal all the determinants of community’s diversity, authors overlooked one important factor –the mean rate of dilution δ (for their system δ given by something like $\log(\text{Th}_{\text{dil}}/\text{CSD})/(\text{average time between consecutive dilutions})$). According to the results of <https://journals.aps.org/prl/abstract/10.1103/PhysRevLett.120.158102> subsequently reproduced in Ref. 43 and cited in the manuscript, higher dilution rates in general tend to reduce the diversity. This could be verified (or disproved) for the system in this study.

> In the revised version, we have added a figure to examine the impact of the dilution scheme. We observe the same trend as the one you have mentioned: higher dilution rates on average reduce the diversity, even though we only observe a modest effect. We have added a supplementary figure (Supplementary Fig 8) to report this trend. Please also see the response to the first comment by Reviewer #1.

4) The exact rules of when and how their turbidostat is being diluted are not properly explained in the manuscript. My understanding is that the system is cyclically diluted (and not periodically, as authors state in the caption to Fig. 2). The dilution happens whenever the overall cell density reaches a predetermined threshold $\text{Th}_{\text{dil}}=10^{10}$ cells/mL (which I had to fish out from the Methods section). The system is then diluted down not by a constant factor but to another predetermined cell density $\text{CSD}=10^4$ cells/mL. This density is referred to as the “initial cell density” on page 19. Is it just the density at the start of the simulation or it reappears in the beginning of every cycle? Authors need to clear up this confusion. It would be also most illuminating if authors were to move an illustration of the cyclic dynamic shown in Fig. S3C to the main text of the manuscript.

> We have revised the manuscript according to your comment to explain the dilution scheme more clearly. You are correct that the setup we are using resembles a turbidostat and is cyclically diluted when the overall population reaches a threshold (now called dilution cell density, Σ_{dil}). At that point, the community is diluted

back to the initial cell density (now called total initial cell density, ΣS_{init}) (Page 6):

“In each simulation, we initially put together several species at equal proportions with a random network of interactions. These communities grow (following Eq 1) from a set total initial cell density (ΣS_{init}) up to a pre-determined threshold cell density for dilution (ΣS_{dil}), upon which the culture is diluted back to the initial cell density (Fig 2A, top).”

Some minor typos: on page 3 “growth promotion influences” -> “growth-promoting influences”.

> We have fixed the typo in the revised version.

We thank all reviewers again for constructive comments.

Babak Momeni and Wenying Shou

REVIEWERS' COMMENTS:

Reviewer #1 (Remarks to the Author):

In my opinion, in this revised version of the manuscript the authors have conveniently addressed the suggestions from the three referees. I find the current version suitable for publication in Nature Communications.

The authors may want to perform a minor correction before publishing the work. During their revision of the work, the authors replaced 'enriched communities' by 'derived communities' in the main text. In some titles in Figures 2 and 4 we can still read 'enriched communities', and maybe they will want to unify these titles with the main text.

Reviewer #2 (Remarks to the Author):

The authors have well addressed all of my comments and I believe this manuscript is now ready for publication.

Reviewer #3 (Remarks to the Author):

I am fully satisfied with the changes made by authors in response to my comments. I enthusiastically recommend the manuscript for publication in Nature Communications.

Response to Reviewers' comments:

We would like to thank the editor and reviewers for their helpful and constructive feedback. We have revised the manuscript accordingly. Below please find the final modifications.

REVIEWERS' COMMENTS:

Reviewer #1 (Remarks to the Author):

In my opinion, in this revised version of the manuscript the authors have conveniently addressed the suggestions from the three referees. I find the current version suitable for publication in Nature Communications.

The authors may want to perform a minor correction before publishing the work. During their revision of the work, the authors replaced 'enriched communities' by 'derived communities' in the main text. In some titles in Figures 2 and 4 we can still read 'enriched communities', and maybe they will want to unify these titles with the main text.

> My apologies for the oversight. The issue is corrected in the revised Figs 2 and 4.

Reviewer #2 (Remarks to the Author):

The authors have well addressed all of my comments and I believe this manuscript is now ready for publication.

Reviewer #3 (Remarks to the Author):

I am fully satisfied with the changes made by authors in response to my comments. I enthusiastically recommend the manuscript for publication in Nature Communications.

> We thank all reviewers again for their constructive contribution to improve our manuscript.